# ScribbleGen: Generative Data Augmentation Improves Scribble-supervised Semantic Segmentation

## Abstract

Recent advances in generative models, such as diffusion models, have made generating high-quality synthetic images widely accessible. Prior works have shown that training on synthetic images improves many perception tasks, such as image classification, object detection, and semantic segmentation. We are the first to explore generative data augmentations for scribble-supervised semantic segmentation. We propose ScribbleGen, a generative data augmentation method that leverages a ControlNet diffusion model conditioned on semantic scribbles to produce high-quality training data. However, naive implementations of generative data augmentations may inadvertently harm the performance of the downstream segmentor rather than improve it. We leverage classifier-free diffusion guidance to enforce class consistency and introduce encode ratios to trade off data diversity for data realism. Using the guidance scale and encode ratio, we can generate a spectrum of high-quality training images. We propose multiple augmentation schemes and find that these schemes significantly impact model performance, especially in the low-data regime. Our framework further reduces the gap between the performance of scribble-supervised segmentation and that of fully-supervised segmentation. We also show that our framework significantly improves segmentation performance on small datasets, even surpassing fully-supervised segmentation. The code and synthetic data will be released.

## 1 Introduction

With the massive leaps forward in modern deep learning, machine learning model capacity has never been higher, with some models reaching billions of parameters (Dehghani et al., 2023; Dosovitskiy et al., 2020). However, for many tasks, the size and complexity of datasets have not kept up with the explosion in model capacity. Since machine learning models perform with a large and rich training dataset, the question of scaling datasets to match model sizes is increasingly pressing. For tasks like Fully-Supervised Semantic Segmentation (FSSS) (Long et al., 2015), however, this is especially expensive due to the need for dense pixel-level annotations. These annotations must often also be produced by experts with domain-specific knowledge, exacerbating the costs of data labeling even further.

Weakly-Supervised Semantic Segmentation (WSSS) seeks to reduce the requirement for dense annotations by using weak annotations. Such methods include scribble-supervised semantic segmentation, where only a fraction of pixels along some lines (scribbles) are provided. However, these methods still lag behind fully-supervised alternatives regarding segmentation quality, with state-of-the-art methods still achieving 2-4% lower mIoU (Wu et al., 2023a; Liang et al., 2022) relative to fully-supervised models.

Another strategy is to produce synthetic training data using image-generative models. Prior works have shown that using Generative Adversarial Networks (GANs) (Goodfellow et al., 2014) to produce training data improves results in image classification (Frid-Adar et al., 2018) and semantic segmentation (Zhang et al., 2021; Bowles et al., 2018), among other tasks. Diffusion models (Sohl-Dickstein et al., 2015; Ho et al., 2020; Song et al., 2021b), a well-known type of generative models, have demonstrated strong performance in terms of controllability (Rombach et al., 2022a; Zhang et al., 2023a) and fidelity (Dhariwal & Nichol, 2021; Saharia et al., 2022). Several studies have successfully applied diffusion models to synthesize training data for image

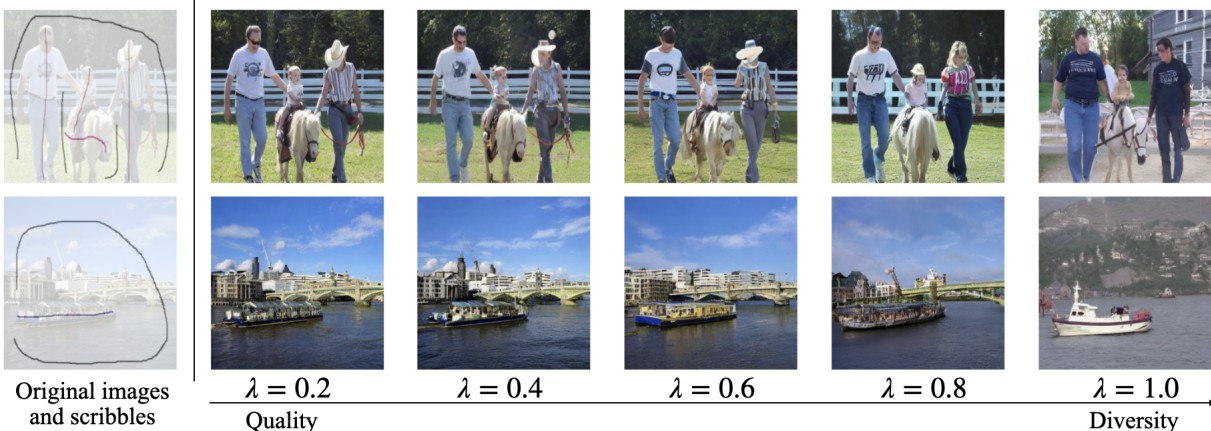

Figure 1: Overview of our ScribbleGen for generating synthetic data for scribble-supervised semantic segmentation. **(Left)** shows the original images overlaid with their scribble annotations. **(Right)** are synthetic data at different encode ratios. These results illustrate the risk of limited data quality under large encode ratios. To resolve this, we propose to trade diversity for quality.

classification (Azizi et al., 2023), object detection (Zhang et al., 2023b), and fully-supervised segmentation (Wu et al., 2023b; Xue et al., 2023). This raises the question: *Can we also leverage the power of diffusion models to synthesize training data to further enhance the performance of scribble-supervised segmentation*?

In this work, we introduce ScribbleGen, a diffusion model conditioned on semantic scribbles to generate high-fidelity synthetic training images. Deep image-generative models such as diffusion models commonly used today, often require large datasets to produce high-quality images (Rombach et al., 2022b). This leads to a paradox where to upscale our training dataset, we need to already have access to a large training dataset. We address this problem by including a new parameter in the generative process, the encode ratio, which trades off image diversity for image photorealism.

Our contributions are summarized as follows:

- To our best knowledge, we are the first to leverage denoising diffusion models for generative data augmentation for scribble-supervised semantic segmentation. Our approach produces a spectrum of synthetic images conditioned on scribbles using different guidance scales and encode ratios.

- We provide detailed analyses and propose several schemes to combine synthetic and real data effectively for scribble-supervised semantic segmentation. We also identify the limitations of naive data augmentation schemes that can harm segmentation performance relative to not using synthetic training data at all.

- We achieve state-of-the-art results in scribble-supervised semantic segmentation, closing the gap between weakly-supervised and fully-supervised models. In particular, our framework significantly improves segmentation results in the low-data regime, where only a limited number of images are available.

## 2 Related work

**Synthetic training data** Numerous efforts have been dedicated to leveraging synthetic data for training perception models. IT-GAN (Zhao & Bilen, 2022) shows that GAN-generated samples can help classification models learn faster and improve performance. DatasetGAN (Zhang et al., 2021) and BigDatasetGAN (Li

et al., 2022) employ GANs (Karras et al., 2019; Brock et al., 2019) for jointly generating synthetic images and their corresponding labels for segmentation tasks.

Recent advances in diffusion models have brought notable stability during training, robust synthesis capabilities (Dhariwal & Nichol, 2021), and enhanced controllability (Zhang et al., 2023a). As a result, there has been a significant shift towards the use of diffusion models for data synthesis, including for image classification (Azizi et al., 2023; Kattakinda et al., 2022), image-text alignment (Fan et al., 2024), object detection (Zhang et al., 2023b; Wang et al., 2024), instance segmentation (Xie et al., 2023a), and semantic segmentation (Nguyen et al., 2023; Wu et al., 2023b; Li et al., 2023; Yang et al., 2023). For example, by fine-tuning an Imagen (Saharia et al., 2022) model on ImageNet (Deng et al., 2009), (Azizi et al., 2023) generates synthetic images from text prompts to improve the performance of image classification. Similarly, D3S (Kattakinda et al., 2022) introduces a novel synthetic dataset specially designed to mitigate the foreground and background biases prevalent in real images. (Nguyen et al., 2023; Wu et al., 2023b;c) jointly generate synthetic images and associated mask annotation, akin to DatasetGAN, using a StableDiffusion (Rombach et al., 2022a) image-generative model. GroundedDiffusion (Li et al., 2023) further generates the triplet of image, mask, and texts to adapt the pretrained diffusion model for open-vocabulary segmentation. FreeMask (Yang et al., 2023) utilizes FreestyleNet (Xue et al., 2023) to synthesize images conditioned on full mask annotations.

Our work diverges from these initiatives by focusing on sparse labels from real images as generative conditions, encouraging the creation of realistic and diverse synthetic images. While FreeMask (Yang et al., 2023) similarly conditions synthetic images on real data annotations, our method uses sparse rather than dense annotations, allowing for broader applications where dense labeling is expensive.

**Guidance in Diffusion models**   Diffusion models excel in various tasks due to their controllability (Zhang et al., 2023a). They're used to generate image content (Ho et al., 2020), image layout (Rombach et al., 2022a; Hu et al., 2023; Meng et al., 2021), audio content (Liu et al., 2023), human motion (Tevet et al., 2022), etc. Guidance signals can also be incorporated to enhance image fidelity (Dhariwal & Nichol, 2021; Ho & Salimans, 2021) relative to unconditional generation. It has been shown that diffusion models can be guided by pretraining a noisy-data-based classifier, known as classifier-guidance (Dhariwal & Nichol, 2021). On the other hand, classifier-free guidance (Ho & Salimans, 2021) removes the need for extra pretraining by randomly dropping out the guidance signal during training. We develop a framework that utilizes classifier-free guidance for generative data augmentation to improve scribble-based segmentation.

Closely related to our work of ScribbleGen is text-guided image inpainting (Xie et al., 2023b; Manukyan et al., 2023), which replaces a specified region in an image with text guidance. However, our work intended for generative data augmentation is different from text-guided image inpainting primarily used for image editing. We condition diffusion model with sparse masks via zero convolution as a variant of ControlNet (Zhang et al., 2023a), while image inpainting methods (Xie et al., 2023b; Manukyan et al., 2023) employ different mechanisms for shape condigining. Besides, we take multi-label semantic mask as condition to generate new images, whereas image inpaiting takes binary mask as condition to edit an image region. We are the first to explore scribble-based semantic image synthesis using diffusion model for weakly-supervised segmentation.

**Weakly-supervised segmentation**   Weakly-supervised segmentation methods use weak annotations rather than full segmentation masks to train segmentation networks for images (Tang et al., 2018; Ke et al., 2021; Chang et al., 2020a; Liang et al., 2022; Wu et al., 2023a; Xu et al., 2022) or point clouds (Unal et al., 2022). Forms of weak annotations include points (Bearman et al., 2016; Tang et al., 2018; Wu et al., 2023a), scribbles (Tang et al., 2018; Liang et al., 2022; Wu et al., 2023a), bounding boxes (Khoreva et al., 2017), image-level tags (Chang et al., 2020a), and text (Xu et al., 2022). These methods can be roughly categorized into two groups. The first group proposes various unsupervised or semi-supervised losses such as entropy loss (Chang et al., 2020b), CRF loss (Tang et al., 2018), and contrastive-learning losses (Ke et al., 2021). The second group iteratively refines full-mask pseudo-labels (Khoreva et al., 2017; Lin et al., 2016) during training to mimic full supervision. Many weakly-supervised approaches rely on class activation maps (CAMs) (Zhou et al., 2016; Chang et al., 2020b) that gives localization cues from classification networks. Our

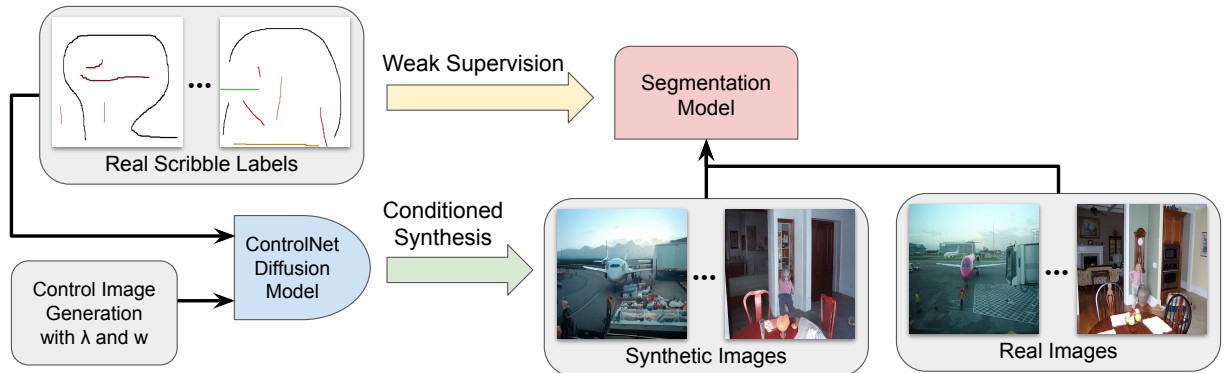

Figure 2: Given a limited number of real scribbles, we pretrain a ControlNet-based diffusion model for high-fidelity synthesis of images conditioned on scribbles. We can control the image synthesis with the encode ratio $\lambda$ and the guidance scale $w$. These image-scribble pairs can then be smoothly integrated into the training of scribble-based semantic segmentation.

generative data augmentation approach complements any existing weakly-supervised segmentation methods, as we show improved performance of several methods with our synthetic data.

Weak annotations can also be provided as input for segmentation networks at test time for interactive segmentation (Xu et al., 2017; Kirillov et al., 2023). For example, Segment Anything (Kirillov et al., 2023) allows prompts including clicks, bounding boxes, masks, or text. While Segment Anything (Kirillov et al., 2023) provides many masks in a semi-automatic way for training interactive segmentation, we focus on synthetic image synthesis for training weakly-supervised segmentation.

## 3 Method

In this section, we describe our method of generative data augmentation for weakly-supervised semantic segmentation outlined in Fig. 2. First, in Sec. 3.1, we provide a background on sampling from diffusion models. Then, in Sec. 3.2, we introduce a variant of ControlNet (Zhang et al., 2023a) conditioned on scribbles and text prompts. We further discuss how to achieve semantically consistent images and trade off diversity and photorealism through guided diffusion and encode ratio in Sec. 3.3 and Sec. 3.4, respectively. Sec. 3.5 proposes several schemes to effectively combine synthetic and real images for training segmentation networks.

### 3.1 Background

Stable Diffusion (Rombach et al., 2022a) is a large-scale pre-trained diffusion generative model that runs the diffusion process in the latent space. The autoencoder is a VQ-GAN (Esser et al., 2021) that quantize images into latent codes. To further improve the quality and fidelity of image systhesis, the classifier-free guidance (Ho & Salimans, 2021) is used by combining the unconditional samples and the conditional samples

$$\tilde{\epsilon}_\theta(\mathbf{x}_t, t;\ \mathbf{c}, w) = (1 + w)\epsilon_\theta(\mathbf{x}_t, t;\ \mathbf{c}) - w\epsilon_\theta(\mathbf{x}_t, t). \tag{1}$$

where $\mathbf{c}$ is a condition and $t$ is a textual prompt. The sampling process is usually DDIM (Song et al., 2021a), which is a deterministic non-Markovian denoising diffusion probabilistic model.

### 3.2 Scribble-conditioned Image Synthesis

We consider a semantic synthesis approach to generating our synthetic training data. The synthetic training data is generated conditioned on real segmentation labels from the training dataset. We leverage a typical denoising diffusion model, ControlNet (Zhang et al., 2023a), to achieve image synthesis conditioned on the segmentation scribbles. Our model is trained using the usual DDPM (Ho et al., 2020) object: given a noisy image $\mathbf{x}_t$ (in reality $\mathbf{x}_t$ is a latent representation as in (Rombach et al., 2022a), but we omit this detail for brevity) and conditioning input $\mathbf{c}$ it predicts the added noise $\boldsymbol{\epsilon}$. Our segmentation scribbles on which the model is conditioned are represented as RGB images in $\mathbb{R}^{h \times w \times 3}$ with different colors for every class, though we explore other representations in Sec. 4.3.

Finally, we note that it is difficult for the ControlNet model to produce semantically consistent images with the given scribble labels. We hypothesize that this is due to the difficulty of encoding class information in RGB images, especially in the early stages of training. Therefore, we supplement our model with text prompts that include all the classes within the image. Adding these prompts significantly improves image class consistency and leads to higher-quality images relative to an unchanging default prompt. We explore the effect of this prompt in Sec. 4.3.

Our ControlNet training objective is thus

$$\mathcal{L}_{\mathrm{CN}}(\boldsymbol{\theta}) = \mathbb{E}_{(\mathbf{x}_{\mathrm{ref}},\mathbf{c}_s,\mathbf{c}_t),t,\boldsymbol{\epsilon}} \left[ \|\boldsymbol{\epsilon} - \boldsymbol{\epsilon}_\theta(\mathbf{x}_t, t, \mathbf{c}_s, \mathbf{c}_t)\|_2^2 \right], \tag{2}$$

where $(\mathbf{x}_{\mathrm{ref}}, \mathbf{c}_s, \mathbf{c}_t)$ is the triplet of the original (unnoised) image, the conditioning scribble label, and the conditioning text prompt and $\boldsymbol{\epsilon}_\theta$ is our ControlNet diffusion model.

### 3.3 Classifier-free Scribble Guidance

We leverage diffusion guidance to further improve semantic consistency between the generated synthetic image and conditional input. Following the proposals from Classifier-free Guided Diffusion (Ho & Salimans, 2021), we randomly drop out 10% of all conditioning scribble inputs $\mathbf{c}_s$, replacing them with a randomly initialized, learned embedding $\tilde{\mathbf{c}}$, when training the ControlNet model. By modifying Eq. 1, we arrive at a new guided noise prediction function:

$$\tilde{\boldsymbol{\epsilon}}_\theta(\mathbf{x}_t, t; \mathbf{c}_s, \mathbf{c}_t, w) = (1 + w)\boldsymbol{\epsilon}_\theta(\mathbf{x}_t, t; \mathbf{c}_s, \mathbf{c}_t) - w\boldsymbol{\epsilon}_\theta(\mathbf{x}_t, t; \tilde{\mathbf{c}}). \tag{3}$$

While ControlNet uses a pre-trained Stable-Diffusion model (Rombach et al., 2022a), which is trained conditionally and unconditionally, scribble drop-out during training can be viewed as finetuning the unconditional diffusion model to our dataset. We have found that the guidance scale, $w$, can significantly impact the quality of generated images, especially with respect to the fine-grain details of the produced image. We further ablate this hyperparameter's impact in Sec 4.3.

### 3.4 Control Image Diversity via Encode Ratio

The vanilla diffusion model denoises sampled Gaussian noise $\mathbf{x}_T \sim \mathcal{N}(0, I)$ iteratively until $\mathbf{x}_0$ at inference time. In practice, synthetic images generated this way may be unrealistic, particularly when training data is limited for our scribble-conditioned diffusion model. To improve photorealism at the cost of diversity, we propose another forward diffusion process parameter, the encode ratio $\lambda \in (0, 1]$. Specifically, we perform $\lambda \cdot T$ noise-adding forward diffusion steps to the input images and, during inference, denoise $\mathbf{x}_{\lambda T}$ iteratively until $\mathbf{x}_0$. Thus, for $\lambda = 1$, there is no change, but for small choices of $\lambda$, there is less noise added to the image $\mathbf{x}_0$. As $\lambda \to 0$, the sampled image will become increasingly similar to the original $\mathbf{x}_{\mathrm{ref}}$. Therefore, a whole spectrum of synthetic images with varying levels of similarity to the reference image can be achieved by varying our choice of $\lambda$. We outline our sampling algorithm, which combines the accelerated DDIM sampling from Sec. 3.1, the scribble guidance from Sec. 3.3, and the encode ratio from Sec 3.4 in Algorithm 1. Fig. 3 shows synthetic images generated with varying guidance scales and encode ratios.

Our approach can be seen as a generalized version of latent inversion (Zhou et al., 2023) and latent prior initialization (Yuan et al., 2024) for which we introduce a new hyper-parameter to adjust the initial noise. Latent prior initialization can be seen as the fixed encode ratio $\lambda = 1.0$ in our experiments.

---

**Algorithm 1** Conditional DDIM sampling with guidance scale $w$ and encode ratio $\lambda$

---

**Require:** $q_\sigma$: forward process;
**Require:** $\mathbf{x}_{\text{ref}}$: a reference image;
**Require:** $w \geq 0$: guidance scale;
**Require:** $\lambda \in [0,1]$: encode ratio;
**Require:** $N \in \{1,\ldots,T\}$: number of reverse diffusion process steps;
**Require:** $c_t$ and $c_s$: text prompt and scribble conditioning;
  $\boldsymbol{\epsilon} \sim \mathcal{N}(0,I)$
  $\tau = [\lfloor \frac{\lambda T}{N} n \rfloor : 0 \leq n \leq N]$                                   ▷ Note if $\lambda = 1$ then $\mathbf{x}_{\tau_N} \sim \mathcal{N}(0,I)$
  $\mathbf{x}_{\tau_N} = \sqrt{\alpha_{\tau_N}}\mathbf{x}_{\text{ref}} + \sqrt{1 - \alpha_{\tau_N}}\boldsymbol{\epsilon}$
  **for** $i = N$ to $1$ **do**                       ▷ Predict added noise using diffusion guidance
    $\tilde{\boldsymbol{\epsilon}}_{\tau_i} = (1+w)\boldsymbol{\epsilon}_\theta^{(\tau_i)}(\mathbf{x}_{\tau_i}, c_t, c_s) - w\boldsymbol{\epsilon}_\theta^{(\tau_i)}(\mathbf{x}_{\tau_i}, c_t, \tilde{c}_s)$         ▷ Accelerated DDIM sampling
    $\hat{\mathbf{x}}_0 = (\mathbf{x}_{\tau_i} - \sqrt{1 - \alpha_{\tau_i}} \cdot \tilde{\boldsymbol{\epsilon}}_{\tau_i})/\sqrt{\alpha_{\tau_i}}$
    **if** $i = 1$ **then** $\mathbf{x}_0 \sim \mathcal{N}(\hat{\mathbf{x}}_0, \sigma_{\tau_1}^2 I)$
    **else** $\mathbf{x}_{\tau_{i-1}} \sim q_\sigma(\mathbf{x}_{\tau_{i-1}}|\mathbf{x}_{\tau_i}, \hat{\mathbf{x}}_0)$     **return** $\mathbf{x}_0$ =0

---

### 3.5 Combine synthetic images with real images

Generative data augmentation can, in principle, produce an infinite amount of synthetic images. However, naively combining real and synthetic images can harm rather than benefit weakly-supervised segmentation models, as we have observed. In particular, it is not clear which choices of the guidance scale $w$ and encode ratio $\lambda$ are optimal. We choose the optimal guidance scale, as determined in Sec. 4.3. For encode ratio $\lambda$, we propose and systematically evaluate two strategies for combining synthetic with real images.

Let $\mathcal{X} = \{\mathbf{x}_1, \ldots, \mathbf{x}_n\}$ denote the set of all real images and $\mathcal{Y} = \{\mathbf{y}_1, \ldots, \mathbf{y}_n\}$ denote the set of all (scribble) labels. Then we produce a set of synthetic images $\hat{\mathcal{X}} = \{\hat{\mathbf{x}}_1, \ldots, \hat{\mathbf{x}}_n\}$ where $\hat{\mathbf{x}}_i = \text{DM}_\theta(\mathbf{y}_i, \mathbf{c}_i; w, \lambda)$ is the output of our trained diffusion model, $\text{DM}_\theta$, conditioned on the scribble $\mathbf{y}_i$ and prompt-condition $\mathbf{c}_i$, given guidance scale $w$ and encode ratio $\lambda$. We may then produce a new, augmented dataset $\mathcal{X}' = \text{concat}(\mathcal{X}, \hat{\mathcal{X}})$ and $\mathcal{Y}' = \text{concat}(\mathcal{Y}, \mathcal{Y})$. Note this means each label, $\mathbf{y}_i$, appears twice in our dataset, once for the real image $\mathbf{x}_i$ and once for the synthetic image $\hat{\mathbf{x}}_i$.

- **Fixed encode ratio** $\lambda$: We choose a fixed encode ratio which gives a fixed synthetic dataset $\hat{\mathcal{X}}$. Using the default value of $\lambda = 1$ yields the most diverse synthetic images with possibly inferior image fidelity. We find the optimal $\lambda$ that gives the best segmentation in our experiments.

- **Adaptive encode ratio** $\lambda$: To avoid hyper-parameter search, we also propose an adaptive scheme for choosing $\lambda$. We gradually increase the encode ratio $\lambda$ while training downstream segmentation networks, similar to curriculum learning. Initially, synthetic images used for training are similar to real images, which are considered an easier curriculum to learn. Synthetic images diverge increasingly from the real images as training progresses. For this case, the synthetic dataset is formed at epoch $e$ as $\hat{\mathcal{X}} = \{\hat{\mathbf{x}}_{1,\lambda_e}, \ldots, \hat{\mathbf{x}}_{1,\lambda_e}\}$ where we follow the encode ratio schedule $[\lambda_1, \ldots, \lambda_E] \in \Lambda^E$ where $E$ is the number of training epochs.

## 4 Experiments

Sec. 4.1 summarize our main results that show improvements on several scribble-supervised segmentation methods using our generative data augmentation. In Sec. 4.2, we further explore the challenging scenario with limited number of real images. We show that naive implementations of generative data augmentation may harm the performance, whereas our data augmentation scheme improves. Sec. 4.3 gives an ablation study on guidance scale and encode ratio, two critical degrees of freedom for our image synthesis.

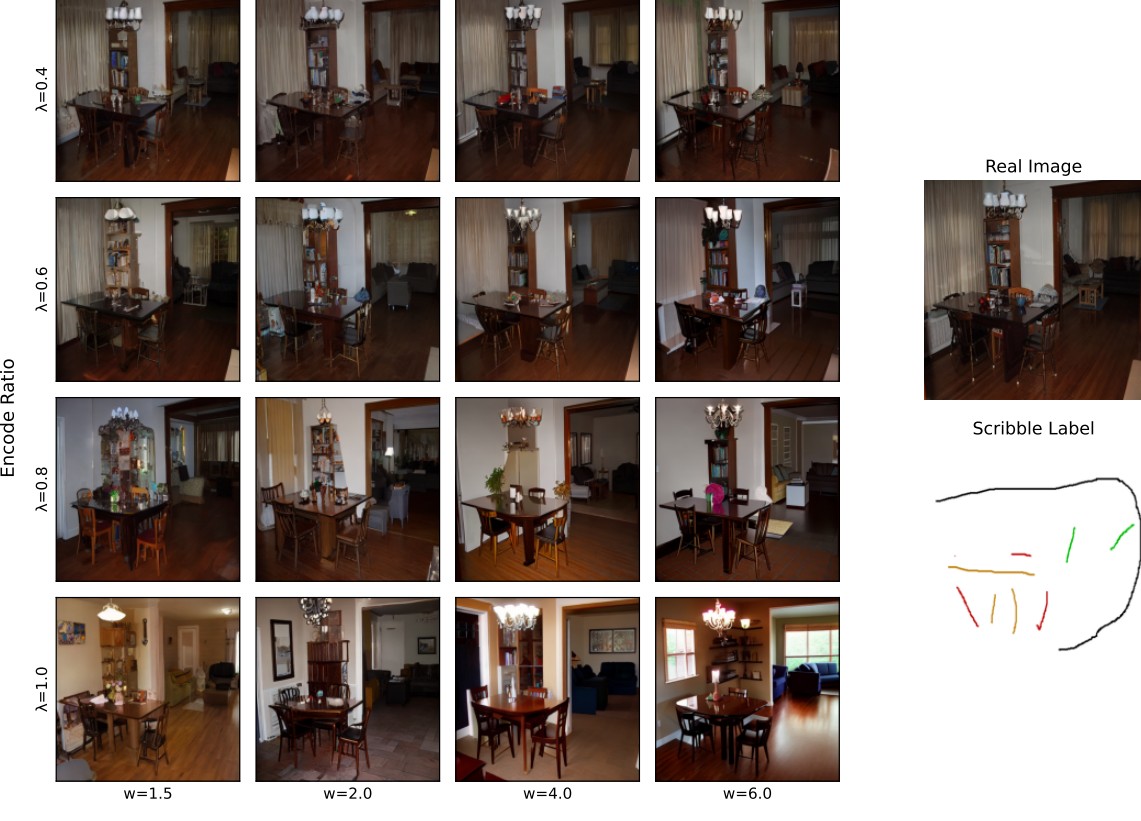

Figure 3: **(Left)**: Our sampled synthetic images conditioned on the ground-truth scribble. By sampling using different guidance scales and encode ratios we are able to generate a whole spectrum of realistic synthetic training images. **(Right)**: The ground-truth real image and corresponding scribble label.

**Dataset and Implementation Details** We report results on the standard PASCAL VOC12 segmentation dataset which contains 10 582 images for training and 1 449 images for validation. We utilize scribbles from ScribbleSup dataset (Lin et al., 2016) with only 3% pixels labeled on average.

For image synthesis, we use a latent diffusion model (Rombach et al., 2022a) with a downsampling rate of $f = 8$, so that an input image of size $512 \times 512$ is downsampled to $64 \times 64$. We use Stable Diffusion 1.5 as the backbone for ControlNet (Zhang et al., 2023a) and finetune ControlNet for 200 epochs with a batch size of 16 using two A100 80GB GPUs. We set $T = 1000$ discrete timesteps for ControlNet and use a linear learning rate scheduler from an initial rate of $10^{-4}$ during training. For scribble conditioning, we randomly dropout 10% of scribbles, replacing them with a learned embedding of the same size. Scribble labels are represented as RBG images in $\{1, \ldots, 255\}^{512 \times 512 \times 3}$. We also provide the text prompt "a high-quality, detailed, and professional image of [list of classes]" as suggested in (Zhang et al., 2023a). We provide visualizations of our synthetic dataset in the supplementary material.

**Evaluation metric.** We evaluate both the diversity and fidelity of the generated images by the Fréchet Inception Distance (FID) (Heusel et al., 2017), as it is the *de facto* metric for the evaluation of generative methods, e.g., (Dhariwal & Nichol, 2021; Karras et al., 2019; Brock et al., 2019; Saharia et al., 2022). It provides a symmetric measure of the distance between two distributions in the feature space of Inception-V3 (Szegedy et al., 2016). We use FID as our primary metric for the sampling quality. We realize, however, that FID should not be the only metric for evaluating the downstream impact of synthetic data for training segmentation networks. Hence, we also report segmentation results trained with synthetic data only to evaluate synthetic data, similar to the Classification Accuracy Score (CAS) proposed by (Ravuri & Vinyals,

2019) but for semantic segmentation. We report the standard mean Intersection Over Union (mIOU) metric for segmentation results.

## 4.1 Generative data augmentation improves scribble-supervised semantic segmentation

For our experiments, we consider two methods of weakly-supervised semantic segmentation, including simple regularized losses (RLoss) (Tang et al., 2018) and the current state-of-the-art in scribble-supervised segmentation, Adaptive Gaussian Mixture Models (AGMM) (Wu et al., 2023a). For both methods, we jointly train them on the original training set and our augmented training set. Both methods also follow a polynomial learning rate scheduler. The sampling of synthetic training images is outlined in Sec. 3.5. Tab. 1 shows improved results using generative data augmentation for both RLoss and AGMM. Our method with synthetic data further reduces the gap between weakly-supervised and fully-supervised segmentation. We show visualizations of our segmentation results with and without using our generative data augmentation in Fig. 6. We also include further visualizations in the appendix.

| Method | Network | Supervision | Synthetic Data | Augmentation Scheme | mIoU (%) |
|---|---|---|---|---|---|
| (1) *DeeplabV3+ (Chen et al., 2018) | MobileNet (Sandler et al., 2018) | Full mask | | – | 72.1 |
| (2) *DeeplabV3+ (Chen et al., 2018) | ResNet101 (He et al., 2016) | Full mask | | – | 79.3 |
| RLoss (Tang et al., 2018) | (1) | Scribble | | – | 68.4 |
| RLoss (Tang et al., 2018) | (1) | Scribble | ✓ | Fixed $\lambda = 1.0$ | 69.4 (+1.0) |
| RLoss (Tang et al., 2018) | (1) | Scribble | ✓ | Fixed $\lambda = 0.5$ | **70.0** (+1.6) |
| RLoss (Tang et al., 2018) | (2) | Scribble | | – | 76.6 |
| RLoss (Tang et al., 2018) | (2) | Scribble | ✓ | Fixed $\lambda = 1.0$ | 76.1 (-0.5) |
| RLoss (Tang et al., 2018) | (2) | Scribble | ✓ | Fixed $\lambda = 0.7$ | **77.0** (+0.4) |
| AGMM (Wu et al., 2023a) | (2) | Scribble | | – | 76.4 |
| *AGMM (Wu et al., 2023a) | (2) | Scribble | | – | 78.1 |
| AGMM (Wu et al., 2023a) | (2) | Scribble | ✓ | Fixed $\lambda = 1.0$ | 78.0 (-0.1) |
| AGMM (Wu et al., 2023a) | (2) | Scribble | ✓ | Adaptive $\lambda$ | 78.7 (+0.6) |
| AGMM (Wu et al., 2023a) | (2) | Scribble | ✓ | Fixed $\lambda = 0.4$ | **78.9** (+0.8) |

Table 1: Generative data augmentation improves scribble-supervised semantic segmentation methods including RLoss (Tang et al., 2018) and AGMM (Wu et al., 2023a) on PascalVOC (Everingham et al., 2010). The best results are shown in **bold**. Numbers in parenthesis are relative improvement / decrease in comparison to the baseline without synthetic data. Note that * AGMM is our re-implementation which gives better results than reported (Wu et al., 2023a).

## 4.2 Low-data Regime Results

For the low-data regime, we only consider the RLoss method due to its simplicity and speed to train. We consider three different reduced datasets with 50%, 25%, and 12.5% of all training images used, respectively. For each of these cases, we train a ControlNet diffusion model on the limited dataset (following the same experimental setup described at the start of Sec. 4) and sample synthetic images as usual. The results of training RLoss on each of the reduced datasets for our different proposed augmentation schemes are reported in Fig. 1.

We notice that the naive data augmentation fails to help in all of our reduced datasets and instead reduces model performance in all but the 50% case. Conversely, our proposed *Adaptive λ Sampling* improves or matches performance for all four datasets. We hypothesize this is due to the lack of training images required to ensure high-quality generation from our diffusion model. This hypothesis is confirmed by the significantly higher FID scores for synthetic datasets generated with limited training data reported in Fig. 5 middle. We also confirm this hypothesis qualitatively in Fig. 4, where we observe that fully synthetic images deteriorate in quality as the number of training images decreases. However, we can stabilize this deterioration by decreasing the encode ratio λ to improve image realism. Using our *Adaptive λ sampling*, the most synthetic (and thus lowest quality) images cannot impact model training as significantly due to the reduced learning from our scheduler.

**Note on evaluation metric** As Figure 5 shows, a lower FID does not necessarily lead to a better mIoU. This observation aligns with a couple of works in training with synthetic data, suggesting FID of synthetic

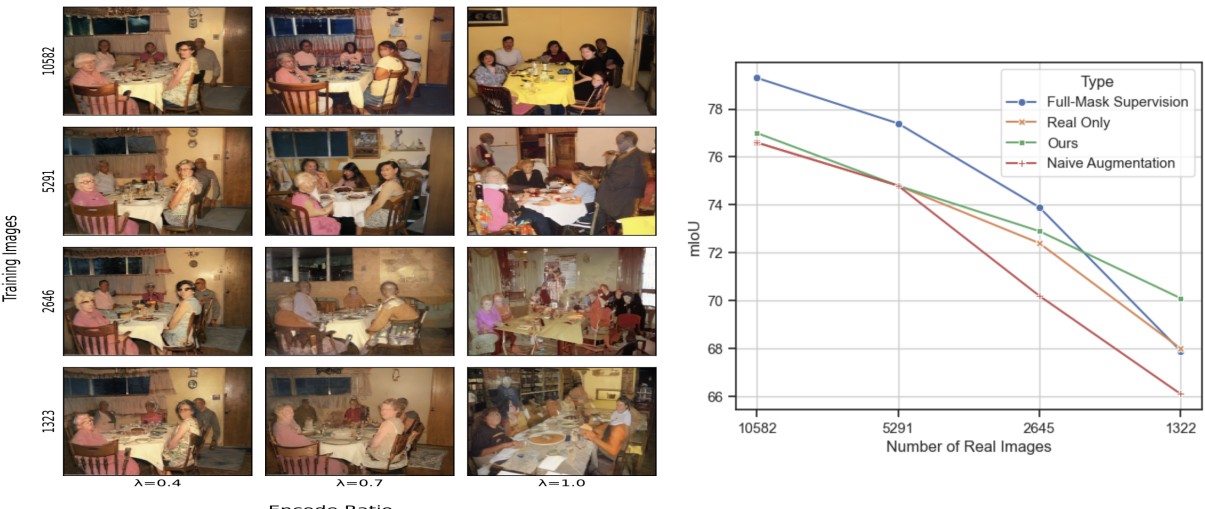

Figure 4: **(Left)**: Synthetic images sampled from diffusion models with different numbers of training images and encode ratios $\lambda$. **(Right)**: Segmentation model performances on PascalVOC and its subsets. All results other than full-mask supervision use scribble supervision with ResNet-based RLoss (Tang et al., 2018) model. Naive data augmentation (i.e., fixed encode ratio $\lambda = 1.0$) harms model performance, especially in the low-data regime, while our augmentation scheme improves performance.

images is not well correlated with the performance of the models trained by those data in downstream tasks (Azizi et al., 2023; Yang et al., 2023; Fan et al., 2024; Wang et al., 2024). To address this problem, Azizi et al. (2023) proposes Classification-Accuracy Score (CAS) as a metric for the quality of synthetic data for image classification; FreeMask (Yang et al., 2023) suggests to use the test mIoU of the segmentation model trained with synthetic images for semantic segmentation; Fan et al. (2024) emphasizes the importance of diversity and recognizability of synthetic images for training CLIP (Radford et al., 2021). In our work, we follow FreeMask (Yang et al., 2023) to use the test mIoU of segmentation model for measuring the quality of synthetic images.

### 4.3 Ablation Studies

**Guidance Scale**   We report the FID scores of our fully synthetic ($\lambda = 1$) datasets as generated by our model trained on all PascalVOC training images in Fig. 5 left. This ablation study is how we decided to use $w = 2$ for all other experiments, as it yields optimal FID. We include further visualizations of the impact of the guidance scale on image synthesis in our supplementary material.

**Encode Ratio**   We report the FID scores of our diffusion models trained with a variable number of images as a function of the encode ratio $\lambda$ in Fig. 5 middle. We observe that the FID increases significantly as the number of training images decreases. However, we can reduce the effect of limited training data by decreasing the encode ratio to promote image realism. This effect is most pronounced for the 1323 image-trained diffusion model, where we reduce the FID score by over 30 points by lowering the encode ratio.

We also evaluate segmentation model performance on synthetic data of varying encode ratios and report the final mIoU in Fig. 5 right. For these experiments, we train segmentation models training using the Fixed $\lambda$ data augmentation proposed in Sec. 3.5 and training exclusively on synthetic training data (i.e., $\mathcal{X}' = \hat{\mathcal{X}}$), akin to CAS (Ravuri & Vinyals, 2019). We observe that the impact of varying the encode ratio $\lambda$ is limited in the data augmentation case but much more significant for the synthetic-only case. We suppose that for the synthetic-only case, the quality of the synthetic images is more important, so decreasing the encode ratio

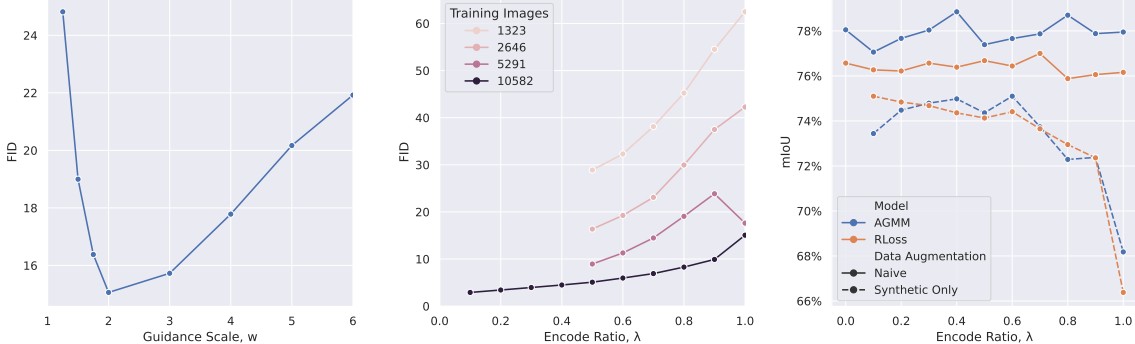

Figure 5: **(Left)**: The FID of our full training dataset when generated with different classifier-free guidance scales. Results are reported for ControlNet trained all 10582 images. **(Middle)**: The FID of our training dataset when generated with different encode ratios. Results are reported for four ControlNet models trained on a different number of images. **(Right)**: The mIoU of a downstream segmentation model when trained on datasets of varying encode ratios. Note $\lambda = 0.0$ corresponds to training on real images only. Results are reported for training on both naive data augmentation and only on synthetic images. In both cases, we use all 10582 images for training.

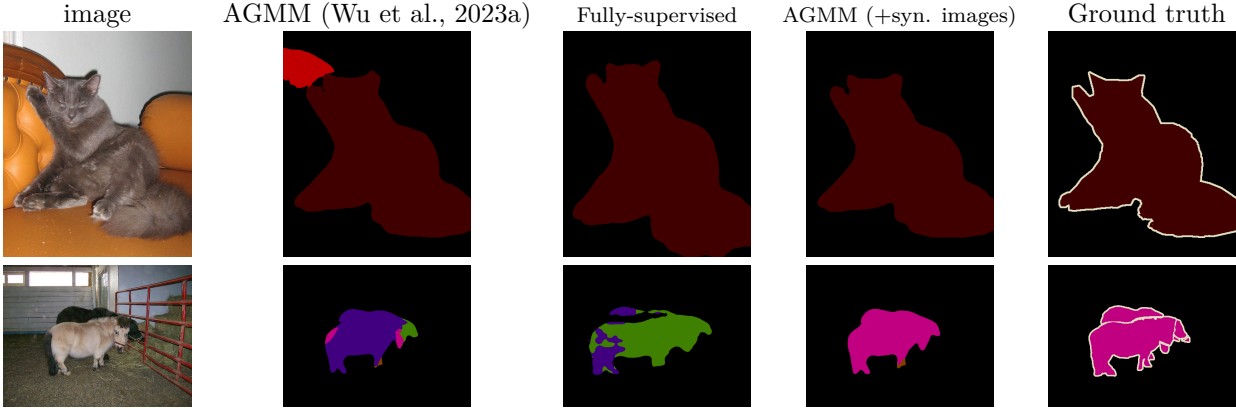

Figure 6: Qualitative results on PASCAL dataset. Our generative data augmentation method improves scribble-supervised semantic segmentation methods such as AGMM (Wu et al., 2023a).

to improve data realism matters more than data diversity. We include further visualizations of the impact of the encode ratio on image synthesis in the appendix.

**Conditioning Input**   We also ablate modifying the conditioning input to ControlNet. We try representing scribble labels as one-hot embeddings in $\{0,1\}^{h \times w \times C}$ where there are $C$ total classes. Using these one-hot embeddings, we obtained a higher FID by 4.4 points relative to RGB embeddings, but we found no improvement in mIoU results using our Fixed $\lambda$ augmentation scheme. We also try using text prompts that don't include the classes in the image. Using unchanging prompts (i.e., "a high-quality, detailed, and professional image") yields lower FID by 3.1 points relative to prompts that include the classes in the image and 1.9% lower mIoU using our Fixed $\lambda$ augmentation scheme.

## 4.4 Comparison to GAN

To show the superior performance of diffusion model over GAN, we compare our ScribbleGen, which is based on diffusion model, with GAN-based SPADE (Park et al., 2019) conditioned on full masks or scribbles. Fig. 7 shows images generated by our method are much better for models trained with the same set of images.

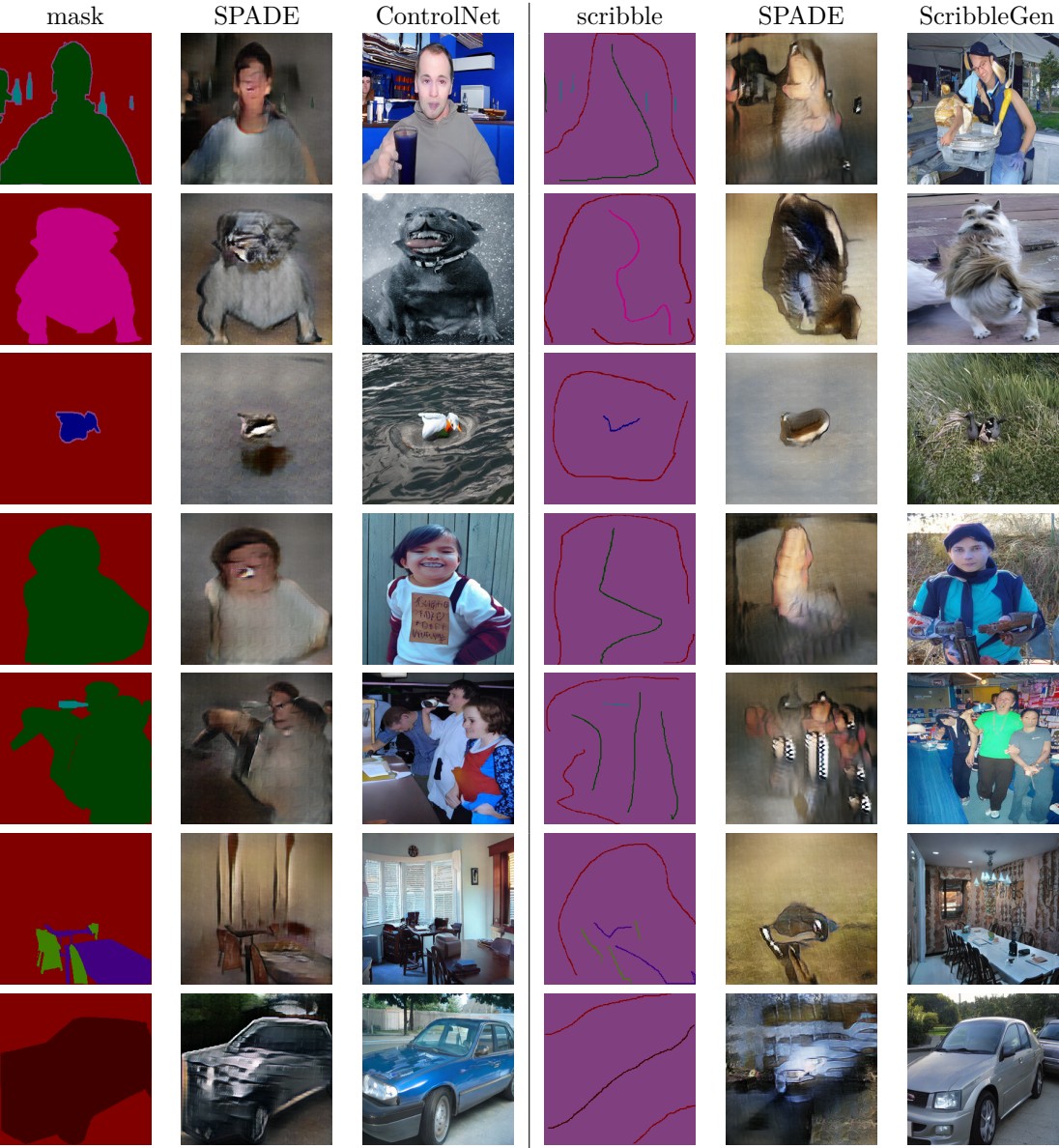

| mask | SPADE | ControlNet | scribble | SPADE | ScribbleGen |
|------|-------|------------|----------|-------|-------------|

Figure 7: Left: Mask-conditioned generation. Right: scribble-conditioned generation with a tailored version of SPADE (Park et al., 2019) and our ScribbleGen. Diffsion model is superior than GAN for both scribble and mask conditioned image generation. We found that diffusion model gives better spatial control and foundation models such as Stable Diffusion are easily accessible. Using synthetic images generated by SPADE, we are not able to improve segmentation with generative data augmentation.

## 5 Conclusion and Future Work

We propose leveraging diffusion models conditioned on scribbles to produce high-quality synthetic training data for scribble-supervised semantic segmentation. We advocate the use of classifier-free guided diffusion and

introduce the encode ratio to control the generative process, allowing us to generate a spectrum of images. We report state-of-the-art performance on scribble-supervised semantic segmentation with our generative data augmentation. In the future, it will be interesting to train generative models for open-vocabulary image synthesis conditioned on sparse annotations. Our generative data augmentation has the potential to improve semi-supervised segmentation. We are also interested in end-to-end training of generative data augmentation and perception models, as metrics like FID are loosely related to perception performances.

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
