# OpenReview forum: "ScribbleGen: Generative Data Augmentation Improves Scribble-supervised Semantic Segmentation"
_TMLR — Withdrawn by Authors_

### Review · Reviewer_dxmv · 2025-02-04

**Summary Of Contributions:**

This paper targets the scribble-supervised semantic segmentation task, which adopts the ControlNet model to generate synthetic images to enlarge the training set size (image only). Two important mechanisms, including the classifier-free scribble guidance and encode ratio-based generation are proposed to respectively improve the semantic consistency and control the diversity of the generated images. Experiments have demonstrated that the proposed ScribbleGen model has achieved state-of-the-art performance on the standard Pascal VOC 2012 dataset, and ablation studies have been conducted to validate the effectiveness of the proposed mechanisms.

**Audience:**

Yes

**Claims And Evidence:**

Yes

**Requested Changes:**

See Cons above

**Strengths And Weaknesses:**

Pros: 1. This paper is well organized. 2. The experiment results seem promising.

Cons: 1. The effectiveness of the different types of control inputs is not ablated, in particular the effect of disabling the scribble condition. 2. Many ablations studies report the FID metric for the synthetic images, instead of the mIoU for synthetic images training only which the authors claim to adopt in the last paragraph of Sec.4.2. 3. I could not find the results related to the first low-data regime result in Fig. 1. 4. Since the diffusion model requires a considerable amount of resources to train, it is better to compare the training cost with other competitive methods, such as TEloss[1], EIL [2], PFA[3], A2GNN[4], etc.

References: [1] Liang Z, Wang T, Zhang X, et al. Tree energy loss: Towards sparsely annotated semantic segmentation[C]//Proceedings of the IEEE/CVF conference on computer vision and pattern recognition. 2022: 16907-16916. [2] Zhou C, Cui Z, Xu C, et al. Exploratory inference learning for scribble supervised semantic segmentation[C]//Proceedings of the AAAI Conference on Artificial Intelligence. 2023, 37(3): 3760-3768. [3] Chan G, Zhang P, Dong H, et al. Scribble-Supervised Semantic Segmentation with Prototype-based Feature Augmentation[C]//Forty-first International Conference on Machine Learning. [4] Zhang B, Xiao J, Jiao J, et al. Affinity attention graph neural network for weakly supervised semantic segmentation[J]. IEEE Transactions on Pattern Analysis and Machine Intelligence, 2021, 44(11): 8082-8096.

---

### Review · Reviewer_zgoo · 2025-02-13

**Summary Of Contributions:**

This paper proposes a novel data augmentation approach using a diffusion model to expand semantic segmentation training datasets by generating additional samples. The method utilises scribble labels as conditioning information alongside the original image and text-based guidance to produce augmented data with controlled variations. Specifically, within a ControlNet framework guided by both textual prompts and visual cues, scribbles encoded as RGB images are leveraged for conditioning.
The main procedure is the fine-tuning of a diffusion model on a given dataset. Subsequently, the model is used to generate augmented samples which one trains on. This approach is orthogonal to traditional sparsely-supervised training methods and provides an additional option for improving segmentation performance alongside those architectures.
The effectiveness of the proposed pipeline is evaluated using FID to measure image fidelity, along with downstream experiments to assess its impact on model performance.

**Audience:**

Yes

**Broader Impact Concerns:**

I have no concerns of that kind.

**Claims And Evidence:**

Yes

**Requested Changes:**

- [**Critical**] Re-run the fully-supervised baseline for ResNet101/DeepLabV3+ or report the DeeplabV3+ metrics from literature for Table1
- [**Critical**] Evaluate the methods using at least one more complex scribble based dataset(e.g. s4Cityscapes, s4ADE20K from the Scribbles4All paper)
- [**Important**] Run the augmentation ablations shown for Rloss and AGMM for an additional SOTA scribble segmentation method
- [**Important**] Update the segmentation method overview in the Related Work Section to include the more recent SOTA methods
- [*Minor*] Fix the citation of DeeplabV3+ in Table1

**Strengths And Weaknesses:**

## Strengths
The paper presents an innovative approach for the improvement of scribble-supervised semantic segmentation through the use of diffusion models for data augmentation. This approach is orthogonal to existing improvements and operates independently of specific segmentation networks, making it broadly applicable. By generating high-quality augmented data, it has the potential to narrow the performance gap between scribble-supervised and fully supervised models. The approach effectively uses state-of-the-art diffusion models to generate samples. Doing so, it further exploits unique properties of scribbles, which provide coarse information about an object's size and orientation without defining precise contours. This allows the diffusion model to introduce meaningful variation while maintaining realism, providing a structured yet flexible way to increase the size of training datasets.

## Weaknesses
- **Baseline Performance Concerns**: The reported baseline results on ResNet101 are significantly lower than those found in other publications (typically around 84.6). If the discrepancy is due to changes in the training procedure, this needs to be explicitly disclosed and justified. Otherwise, this erroneous value might mislead readers.
- **Outdated Experimental Models**:  RLoss, a method cited and used for comparison, is seven years old and no longer represents the state of the art. There are more modern scribble segmentation frameworks that should be used. (For example TEL(arxiv.org/abs/2203.10739) that is also a method based on a regularized loss)
- **AGMM Performance and Missing Comparisons**: While AGMM is recognized as a state-of-the-art approach, the paper does not sufficiently discuss its lack of generalization to other datasets. Recent work has shown that ScribbleSup is an outdated dataset and that methods which perform well on it might not generalise well to more complex datasets(Scribbles for All, proceedings.neurips.cc/paper_files/paper/2024/file/51cd2b3608d7ae17a9fadcc6e1f68629-Paper-Datasets_and_Benchmarks_Track.pdf). Additionally, other modern SOTA methods are missing from the comparisons, despite their relevance since AGMM is a method that does not perform well on other datasets.
- **Incorrect Claim**: While the incorrectly reported baseline does not impact the demonstrated improvements of the data augmentation method on the two reference scribble segmentation models, it contradicts the paper's claim of narrowing the gap between scribble-supervised and fully supervised semantic segmentation. SOTA segmentation methods omitted in this paper outperform the best results presented in this paper.
- **State-of-the-Art Methods Outperform the Proposed Model**: Existing state-of-the-art methods achieve better results than the best-performing model presented in the paper. Furthermore, the performance gap between using the proposed augmentation and not using it appears to decrease when applying stronger segmentation methods and backbones.
- **Outdated Baselines and Literature Review**: The paper does not include recent segmentation methods such as SASFormer(arxiv.org/abs/2212.02019) and TEL. Overall, the literature review lacks references to scribble segmentation approaches developed after 2020.
- **Incorrect or Incomplete Citations**: The paper incorrectly cites DeepLabV3+ by pointing to the inital DeepLab paper in the experimental section (Table 1).

While the weaknesses outlined here are partially severe, it appears like they could be easily alleviated without changes to the core method of the paper.

---

### Review · Reviewer_bmji · 2025-03-01

**Summary Of Contributions:**

This paper proposes a data-augmentation method using Stable Diffusion for scribble-supervised semantic segmentation. It identifies two key factors, guidance scale and encode ratio, in balancing the combination of conditional and unconditional samples and controlling the noise-adding forward diffusion steps, respectively. It claims that these factors significantly impact the fidelity and diversity of the synthesized images. The paper presents an experimental analysis on PASCAL VOC, demonstrating the effectiveness of the proposed approach, particularly in low-data regimes.

**Audience:**

Yes

**Broader Impact Concerns:**

The proposed work does not raise significant ethical concerns.

**Claims And Evidence:**

Yes

**Requested Changes:**

1. Provide a more detailed theoretical discussion or an intuitive explanation of why these two parameters are critical.
2. Provide clearer guidelines on the optimal values of guidance scale and encode ratio.
3. Expand the experiments by including more datasets, different segmentation settings, and comparisons with state-of-the-art methods.

**Strengths And Weaknesses:**

**Strengths**:
1. The paper explores an interesting direction by leveraging diffusion models for scribble-supervised segmentation.
2. The experimental analysis provides insights into how diffusion parameters influence the generated samples.
3. The proposed method demonstrates improvements over baseline methods, suggesting its practical utility in data-scarce settings.

**Limitations**:
1. Lack of Theoretical and Intuitive Justification: (**i**) While Stable Diffusion is a well-known model, the paper lacks a clear theoretical or intuitive explanation of why the two identified factors (guidance scale and encode ratio) are particularly relevant in this context. (**ii**) The experimental analysis does not provide sufficient validation regarding the optimal choice of these parameters across different scenarios. The selection of guidance scale appears empirical, without a well-grounded conclusion on how different values affect segmentation performance in varying conditions.

2. Limited Experimental Scope and Comparisons: (**i**) Table 1 presents only marginal improvements over baselines on a single dataset and setting, without comparisons to state-of-the-art methods. A broader evaluation, including different datasets or settings, would strengthen the claims. (**ii**) Prior works [1,2,3] evaluated their methods under additional settings such as scribble-shrink and scribble-drop or other datasets, which are missing in this study. Including these settings would allow a more comprehensive assessment of the method’s robustness.

[1] Scribble hides class: Promoting scribble-based weakly-supervised semantic segmentation with its class label (AAAI 2024)

[2] CC4S: Encouraging Certainty and Consistency in Scribble-Supervised Semantic Segmentation (PAMI 2024)

[3] Scribbles for All: Benchmarking Scribble Supervised Segmentation Across Datasets (NeurIPS 2024)

---

### Note · Authors · 2025-03-02

**Comment:**

Dear reviewers and editors,

We appreciate the thorough, timely, and critical review from your prestigious venue.

After reading the reviews carefully, we decided to withdraw the paper. We feel this paper still has many gaps to fill in terms of theory and experiments, as pointed out by the reviewers. We think our work needs to be substantially improved to become impactful for the community. We will keep working relentlessly on the important problem of generative data augmentation.

Thanks again for your time in giving feedback!
-Authors

**Withdrawal Confirmation:**

I have read and agree with the venue's withdrawal policy on behalf of myself and my co-authors.